# Flowering and Seed Production across the Lemnaceae

**DOI:** 10.3390/ijms22052733

**Published:** 2021-03-08

**Authors:** Paul Fourounjian, Janet Slovin, Joachim Messing

**Affiliations:** 1Waksman Institute of Microbiology, Rutgers University, Piscataway, NJ 08854, USA; 2Genetic Improvement of Fruits & Vegetables Laboratory, USDA, Beltsville, MD 20705, USA; janet.slovin@usda.gov

**Keywords:** duckweed, *Spirodela polyrhiza*, *Lemna gibba*, *Lemna minor*, *Wolffia microscopica*, flowering protocols

## Abstract

Plants in the family Lemnaceae are aquatic monocots and the smallest, simplest, and fastest growing angiosperms. Their small size, the smallest family member is 0.5 mm and the largest is 2.0 cm, as well as their diverse morphologies make these plants ideal for laboratory studies. Their rapid growth rate is partially due to the family’s neotenous lifestyle, where instead of maturing and producing flowers, the plants remain in a juvenile state and continuously bud asexually. Maturation and flowering in the wild are rare in most family members. To promote further research on these unique plants, we have optimized laboratory flowering protocols for 3 of the 5 genera: *Spirodela*; *Lemna*; and *Wolffia* in the Lemnaceae. Duckweeds were widely used in the past for research on flowering, hormone and amino acid biosynthesis, the photosynthetic apparatus, and phytoremediation due to their aqueous lifestyle and ease of aseptic culture. There is a recent renaissance in interest in growing these plants as non-lignified biomass sources for fuel production, and as a resource-efficient complete protein source. The genome sequences of several Lemnaceae family members have become available, providing a foundation for genetic improvement of these plants as crops. The protocols for maximizing flowering described herein are based on screens testing daylength, a variety of media, supplementation with salicylic acid or ethylenediamine-*N*,*N*′-bis(2-hydroxyphenylacetic acid) (EDDHA), as well as various culture vessels for effects on flowering of verified Lemnaceae strains available from the Rutgers Duckweed Stock Cooperative.

## 1. Introduction

The five genera and 36 species comprising the Lemnaceae family, commonly known as duckweeds, are the smallest, fastest growing, most morphologically reduced, and widely distributed family of angiosperms [1,2,3]. These aquatic monocots are found floating and rapidly clonally dividing on still, nutrient rich, waters worldwide. Duckweeds are rare among plants for being a complete protein source, with an amino acid content similar to eggs [4,5]. Wild or greenhouse grown *Wolffia* species are an especially resource-efficient food source. Protein concentrates from *Lemna* species are being explored as a scalable and economical way to meet the rapidly increasing demand for plant-based proteins [6,7]. Members of the Lemnaceae are promising industrial crops that can grow at rates of 13–38 dry tons/hectare per year, compared to 6–9 and 2.5–3.5 dry tons/hectare per year for maize and soybean, respectively, reported by the FAO [8,9]. Duckweeds can grow on non-arable land in agricultural or industrial wastewater, recapturing nutrients instead of needing energy intensive fertilizers, without pesticides, while cleaning the water and producing a biomass that can be used as animal feed or biofuel [6,7,8]. Research into this family’s unique biology will result in improving the ways we use these distinctive plants to sustainably provide clean water, food, and energy in the future.

The Lemnaceae plant body generally consists of a small (0.5 mm–2 cm), flat leaf-like structure called a frond. Plants in the *Spirodela*, *Landoltia*, and *Lemna* genera have rhizoid structures, while the smaller, simpler, and more recently evolved *Wolffiella* and *Wolffia* genera lack these structures [1]. *Wolffia microscopica*, one of the smallest flowering plants, has a rootlike projection described in detail in Sree et al., 2015 [10], the function of which remains unknown. Many Lemnaceae species can overwinter in temperate climates by asexually budding a starch rich modified frond known as a turion to create an asexual organ of perenniation. The turion sinks to the bottom of the pond, then rises as a frond in the spring [1,3,11].

Due to their small size, rapid asexual and clonal growth, ease of aseptic cultivation, and simple morphology, the Lemnaceae were, for many years in the past, used as model systems for studying a wide array of biological and biochemical processes such as flowering [12,13,14], hormone and amino acid biosynthesis [15,16,17], the role of the D1 protein and regulation of expression of chlorophyll binding protein genes in photosynthesis [18,19], and many other aspects of plant biology [1,20,21].

In addition, their aquatic lifestyle makes duckweeds particularly useful for phytotoxicity testing, and standardized experimental protocols using duckweeds were developed for testing water quality [22,23,24]. To facilitate rapid and accurate measurement of duckweed growth in toxicity tests, there has been a recent shift from frond counts and fresh or dry weight measurements to image analysis software, which can measure total area, plant health, and possibly flowering. The company Lemnatec sells commercial imaging chambers, with software packages, while researchers have applied the Aphelion software package [25], the NI vision Assistant software [26] in data analysis pipelines, or the free software Image J, with manual assistance in selecting green surface area (P. Fourounjian, unpublished) to take non-destructive measurements of biomass.

Hundreds of diverse duckweed strains from around the world were collected by Elias Landolt [27], and many of these are now curated and stored at the Rutgers Duckweed Stock Cooperative, the RDSC (http://www.ruduckweed.org/ (accessed on 26 January 2021)) Additional collections are found worldwide (listed on the RDSC website), making the morphologically diverse germplasm easily available to researchers. The Landolt strains have retained the original 4 digit identifiers, while newer strains have been assigned a 3 digit code, for example, DWC130. While distinguishing the 36 species based on morphology is challenging, they can be easily identified by genotyping or barcoding at precise intergenic spacers [28].

Renewed interests in the commercialization of members of the Lemnaceae have spurred the publication of draft genomes of: *S. polyrhiza* strains 7498 [29] and 9509 [30]; *L. minor* strains 5500 [31] and 8627; *L. gibba* G3 strain 7741; and *W. australiana* strain 8730 (https://www.lemna.org/ (accessed on 26 January 2021)). The *Landoltia punctata* transcriptome was recently described, so 4 of the 5 genera are now under genomic scrutiny [32,33]. A transcriptomic study of turion formation in *S. polyrhiza* [11]; two maps of miRNAs in *S. polyrhiza* with cleaved targets, [30,34]; and *Spirodela* proteome analysis [35] together with several other genomic mapping advances [36,37,38], have resulted in *S. polyrhiza* strains 7498 and 9509 having the best characterized genomes in the family. The discovery that *Spirodela* has the least RNA directed DNA methylation (RdDM) of any plant studied to date, yet surprisingly few transposons [29,30,39] aptly illustrates the importance of developing resources with which to explore a wide diversity of plants.

There are 19 published protocols for stable transformation and transient gene expression in Lemnaceae, allowing for genome manipulation in 8 species, in all genera except *Wolffiella* [40]. In addition, the metagenome, or microbiome of these plants can be readily studied and manipulated. Plant growth promoting bacteria found in natural Lemnaceae microbiomes are being investigated as a means to enhance growth and ability to purify wastewater. As wild duckweeds can be surface sterilized, grown aseptically, and cultures then re-inoculated with a chosen microbe or intentionally engineered microbial communities, the duckweeds are excellent models to study interactions between plant and bacteria, or fungi, and even algae [41,42].

Members of the Lemnaceae grow in a juvenile and asexual manner indefinitely, and then produce flowers in a matter of days if exposed to the correct stimuli. This allows precise study of individual stimuli or genetic pathways which initiate floral development, rather than the earlier or later flowering phenotypes observed in *Arabidopsis* and other commonly studied annual plants, where regulation of flowering is complicated by multiple active pathways [43,44]. The Lemnaceae were used to search for the mythical “florigen”, first hypothesized in 1937 [45]. Landolt and Kandeler’s multivolume monograph, The Family of *Lemnaceae*—a monographic study Volumes 1 and 2 [1,3] and a later review by Pieterse [46] describe 112 studies to investigate factors affecting floral regulation using duckweed. Day-length was paramount in every study, while salicylic acid (SA), a plant hormone now understood as important for abiotic stress response and pathogen resistance, was the most studied chemical inducer. Chelating agents such as ethylenediamine-*N*,*N*′-bis(2-hydroxyphenylacetic acid) (EDDHA), were also frequent players in those studies. 

While many studies of the duckweeds describe ways to induce flowering in the laboratory, there are only two reports of seed production in the laboratory. This is important as the production of seed underlies establishment of breeding protocols to improve agricultural traits. In the 1950s, Maheshwari described the anatomy of *W. microscopica* and *L. paucicostata* (now *L. aequinoctialis*) seed collected from the wild, showing the presence of a cellular endosperm [47]. A study of IAA accumulation in a tissue culture induced large mutant line of *L. gibba* G3 [16] described that gentle shaking of flowering plants growing on liquid “E” medium resulted in production of seed, which fell to the bottom of the culture flask and germinated when placed in fresh media. Recently, Fu et al., 2017 [48] were the first to produce hybrid duckweed seed by manually crossing under a dissecting microscope, two strains of *L. gibba* G3 flowering on modified Hoagland’s (MH) medium containing 20 μM SA. In that study, only one of the strains used in the cross, 7741, was able to produce viable pollen. Strain 7741 was manually crossed to a male sterile strain to produce the first hybrid strain created in a laboratory. 

The long history of research using duckweeds and the rapidly increasing omics resources available make these plants attractive for use in exploring basic biological questions related to the neotenous, aqueous lifestyle, while simultaneously accelerating the domestication of the fastest growing plants into a sustainable crop species. Yet, one could argue that no duckweed species would ever be a “model organism” unless there are robust flowering protocols and the ability to produce seed. We describe here protocols for reliably obtaining flowering cultures of three genera, that will enable development and floral regulation studies, and facilitate development of protocols for production of improved varieties, whether by classical breeding or gene editing. Such protocols are fundamental to performing the mutant screens, genetic experiments and genome wide association studies that have facilitated research in other model plants, and the commercial breeding of other crops.

## 2. Results

### 2.1. Spirodela Polyrhiza

Initial screening experiments involved three media, various SA or EDDHA concentrations and combinations, and several light regimes. Guided by previous work on other *Spirodela* strains [49,50], we initially screened 3 media: E; Hoagland’s (Hg); and Schenk Hildebrandt (SH) (Appendix A) with and without SA and EDDHA over 28 days in 16 h light:8 h dark long days (LD), or continuous light (CL) for induction of flowering (Appendix A). Flowering was very sparse or completely absent for both genomically characterized strains 7498 and 9509 on any medium in LD. Neither strain flowered on E (pH 4.6), or E at pH 5.8, although a senescent culture of 9509 flowered on E supplemented with 25 μM EDDHA. In these screens, the highest flowering rate, 4%, was seen with strain 7498 on SH and Hg, supplemented with 1.5 or 2.0 μM SA under CL. By day 14, strain 9509 was senescing on SH, so screening was continued with Hg only, and all later experiments were performed using Hg. A second screen was performed to test whether using different culture vessels affected flowering. Strain 7498 exhibited higher flowering rates than 9509 in any culture vessel. Cultures in Petri dishes and larger flasks marginally outperformed those in small flasks. In this set of screening conditions, the optimal SA concentrations were 1.0 μM and 1.5 μM (Appendix A). 

Although a low red/far-red light ratio accelerates the flowering response in *Arabidopsis*, [51], screens with cultures growing with supplemental far-red light failed to produce flowers (Appendix A). Shifting cultures of strain 7498 from LD or CL to Petri dishes under 12 h light:12 h dark led to 9% of fronds flowering with some dehiscent anthers (Appendix A). Replicated testing of strain 7498 with these conditions showed that visible flowering started at day 11, and peaked at day 14 with a maximum rate of 6% independent of whether the inoculant was grown in LD or CL (Appendix A).

Using 4 week old cultures as a source of inoculant improved flowering rate. In Hg medium, Petri dishes of strain 7498 inoculated with three 3–5 frond colonies from one month old cultures growing in Hg medium, began flowering after 10 days in CL (Figure 1). By day 24, strain 7498 had a significantly higher flowering rate than strain 9509 the 1.5 μM SA condition (*p* < 0.01). By day 28, when the culture filled the dish, 11 ± 1% of the 7498 fronds growing on media containing 1.5 μM SA were flowering, significantly different from 2.0 μM SA (*p* < 0.01). Under the same conditions, only 1% of strain 9509 fronds were flowering (Figure 1). Both strains produced flowers with pistils secreting stigmatic fluid as well as flowers with dehiscent anthers, although 7498 did so at a higher rate (Figure 2).

Occasionally, abnormal flowers were seen in Petri dish cultures. Unusual kidney shaped fronds of strain 7498 produced flowers that did not appear to emerge from the normal location on the frond (Figure 3A). Strain 9509 occasionally produced flowers with what appeared to be the purple ventral side of the frond overgrowing and covering the anthers (Figure 3B). On this frond, a rhizoid, which would normally grow from the ventral surface of the frond, is growing vertically up out of the medium. Completely detached, aborted developing pistils and anthers enclosed in the spathe were found floating in strain 9509 cultures (Figure 3C).

Both strains of *Spirodela* produced some viable pollen according to the Alexander’s stain [52], that was capable of germinating in vitro (Figure 4). In strain 7498, 69% or 161 of 234 total pollen grains from dehiscing anthers stained as viable, and 48% (47 of 97 total) germinated and produced pollen tubes. Only 26% (207 of 803) of the pollen from strain 9509 dehiscing anthers stained as viable, and only 9.7% (10 of 103) produced pollen tubes. 

On average, strain 7498 formed 518 ± 98 turions per flask over a 2 month culture period, and an average of 309 ± 66 turions were found per flask of strain 9509 over the same time period. After 2 months, no fruit or seed were found at the bottom of flasks or plates of either strain. Gentle rotary shaking of both strains, and manually self-pollinating 6 fluid secreting pistils of 7498 with dehiscent anthers failed to result in seed formation.

### 2.2. Wolffia Microscopica

A screen in which plants were cultured on Hg, with daylength, chelating agent EDDHA, and sucrose addition varied, clearly showed that LD were better for flowering than CL, provided that sucrose and EDDHA were present (Appendix A). EDDHA enhanced frond multiplication when sucrose is present, although plants were capable of multiplying without either sucrose or EDDHA. However, cultures without sucrose and EDDHA senesced earlier than those with sucrose but without EDDHA. Overall, the screen indicated that in LD, Hg supplemented with sucrose and either 25 or 75 μM of EDDHA resulted in the highest flowering rates (Appendix A).

Further screens with E and Hg media in flasks and Petri dishes exhibited very high variability, but overall greater flowering was seen with E, and on this medium no influence of EDDHA on flowering rate was observed (Appendix A). This screen indicated that when growing on E, *W. microscopica* was greatly affected by the type of culture vessel. 

No flowering was observed if *Wolffia* was grown in cotton stoppered flasks containing 100 mL of E, whereas if the plants were grown in 10 mL of E in a 6-well plate, 38 ± 2% of the fronds were flowering over the same time period (Figure 5). After day 7, flowering was significantly greater in parafilm sealed 6-well plates (red) than in parafilm sealed Petri dishes (green) (*p* < 0.05) and no significant differences were found between 6-well plates sealed or not sealed with parafilm (Figure 5). Under these conditions, the abundant pistils were easily observed due to a drop of stigmatic fluid, which was almost the size of the frond, and dehiscent anthers were also abundant (Figure 6). Pollen from these anthers stained as 56% (94 of 169) viable, and 28% of the pollen grains formed germination tubes (65 of 235) (Figure 6). Despite plentiful flowering, none of the cultures, either gently shaken or stationary, were observed to produce seed.

### 2.3. Lemna

No flowering was observed in our initial experiments with DWC130, in contrast to what was reported for this strain in Slovin and Cohen 1988 [16]. As of 2017, there were 5 strains of *Lemna gibba* G3 listed in the RDSC inventory: DWC114 (7741), collected in Sicily, which is the pollen producing strain described by Fu et al. [48], DWC130 and DWC131, labeled as a self-fertile parental line and a larger auxin mutant jsR1 regenerated from tissue culture [16], DWC132, which was contributed by Biolex Corporation, and DWC128 from the Waksman Collection, which was not included in this study. Contrary to expectations, DWC114 was, in fact, the largest of the 4 strains we grew. DNA barcoding using chloroplast atpH-F primers [28] unequivocally showed that strain DWC114 is *L. gibba* G3, and revealed that the other 3 strains are, in fact, *L. minor* (Table 1). The DWC114 barcode sequence is an exact match to both *L. gibba* 5504 and *L. gibba* 7741 described by Fu et al. [48]. DWC132 was further barcoded using matK primers, which clarified its identity as *L. minor*. Barcoding revealed a high degree of similarity among the three *L. minor* strains, DWC130, 131, and 132, with all closely matching *L. minor* strain 7210. The auxin mutant jsR1 appears to have been lost in one of the collection’s multiple transfers, while DWC114 appears to be the *L. gibba* G3 originally described in Hillman 1961 [53]. Based on the results, our colleagues at the RDSC updated the database as part of their curation of the global collection.

As preliminary screening for flowering rate, all 4 strains were grown in flasks of E with 0–100 μM SA in CL, and also in E with either 20 μM SA or 20 μM SA + 75 μM EDDHA in either LD or CL (Appendix A). Overall, this screen showed that DWC114 had the highest flowering rate of the 4 *Lemna*, that 24 h of light promoted flowering, that the optimum SA concentration for flowering was 20 μM, and that EDDHA had no obvious additive effect on flowering. 

Under these conditions, *L. gibba* DWC114 produced perfect flowers, with a translucent pistil often topped with a drop of stigmatic fluid and two dehiscent anthers (Figure 7A). *L. minor* DWC130 (Figure 7B) produced small pistils observable at 16× magnification with no stigmatic fluid, but no anthers. *L. minor* strains DWC131 (Figure 7C) and DWC132 (Figure 7D) produced large pistils and large drops of stigmatic fluid, with two white but non-dehiscent anthers which were prone to detaching and sinking to the bottom of the flask. 

To determine if flowering of *L. gibba* G3, DWC114, was affected by growth in constant light, plants were cultured, in triplicate, in cotton stoppered flasks containing E supplemented with no SA or 20 μM SA, in LD or CL (Figure 7E). The results show that the presence of SA was required to produce flowering fronds until very late in the culture cycle. In the presence of SA, flowering under CL was significantly greater than in LD at day 14 (*p* < 0.01) but the appearance of flowers was essentially equal thereafter under either light condition until the end of the culture cycle.

The production of viable pollen by DWC114, DWC131, and DWC132 was assessed with modified Alexander’s stain [52] (Figure 8A,B) and by pollen germination assays (Figure 8C). Pollen grains from dehisced anthers of *L. gibba* G3 DWC114 were largely viable as assessed by staining (83% viability) (Figure 8(A1,A2)), with 43% capable of forming a pollen tube (59 of 131) (Figure 8(A3)). The non-dehiscent anthers of *L. minor* strains DWC131 and DWC132 contained mostly non-viable pollen, with 1% (3 of 227), and 2% (11 of 454), respectively, of pollen grains staining as viable (Figure 8(B1,B2,C1,C2)). None of the pollen from these anthers produced germination tubes (Figure 8(B3,C3)).

The effect of SA on seed production by *L. gibba* G3, DWC114 (Figure 9) was determined under two light regimes and at four concentrations of SA (10–30 μM SA) over a 40 day culture period. Although similar rates of flowering were observed in LD if SA was present, after 25 days in culture flowering percent dropped to zero by 40 days if SA was not present (Figure 9A). In contrast, the number of seed produced remained low until 25 day of culture, and increased thereafter until 40 days, independent of whether SA was present or not (Figure 9B). In LD, plants growing on E with 30 μM SA produced the fewest seed, an average of 25 ± 0.3 seed per flask. Under CL conditions, flowering remained high throughout the 40 day culture period with small increases attributable to the presence of SA (Figure 9C). In comparison to cultures growing in LD, seed production by plants in CL was substantially higher, with cultures containing 30 μM SA producing almost 150 seed per flask by day 35 (Figure 9D).

Additional media supplements, such as yeast extract and bactopeptone (Eye) with 75 μM EDDHA, were tested for their reported ability to enhance seed production [16]. Addition of 20 μM SA slightly improved flowering (Appendix A), however, seed production was decreased substantially with SA in the enriched medium (Appendix A). In comparison, plants growing on Eye + 75 μM EDDHA produced fewer seed than plants growing on E supplemented with 30 μM SA.

Developing seed of *L. gibba* G3 DWC114 were visible within the frond due to the highly pigmented operculum at one end (Figure 10A). Occasionally, developing seed within the ovary (Figure 10B) were seen detached from the mother frond, although most seed became apparent when they became free of the ovary and dropped to the bottom of the flask (Figure 10B). Seeds could then be harvested using a Pasteur pipette and stored dry, or in the cold in diluted E. Seeds germinated if transferred to fresh E (Figure 10C), achieving a germination rate by day 3 of 65% or higher, independent of LD or CL light conditions (Figure 10). 

## 3. Discussion

The majority of the vast body of flowering research using members of the Lemnaceae was conducted before the sequencing of the human and *Arabidopsis* genomes [54,55,56] and before the first miRNA was discovered in *Caenorhabditis* [57] or miRNAs were found in *Arabidopsis* [58]. Today’s technology, which now allows for determination of the genetic mechanisms involved in flowering in these neotenic aquatic monocots, is still dependent on being able to have a convenient and reliable means of inducing flowering in the laboratory. While previous publications on flowering almost always focused on a single genus of the Lemnaceae, we developed protocols for obtaining flowering dependably in 3 different genera, and applied lessons between them, as a means of enabling the further study of the genetic mechanisms regulating floral induction and progressing toward breeding protocols in this plant family.

The first step in developing reliable flowering protocols was the standardization of the number and the birth order of the inoculating fronds. In *Lemna*, first daughter fronds are more robust that those produced subsequently [59,60,61]. Birth order also influences production of turions by *S. polyrhiza* [62]. We consistently used 4-frond colonies of *Lemna* and 3–5 frond colonies of *Spirodela* as inoculant to help standardize growth and flowering rate. In this way, a colony would have a grandmother frond, with an attached mother frond, and a developing daughter frond, which would be the first daughter. Depending on environmental conditions, a first daughter frond of *Lemna gibba* is capable of producing 12–14 daughters of its own before senescing, although the production of a flower significantly reduces the number of possible daughter fronds [63]. 

It is, therefore, worth considering that while the number of daughter fronds produced by a particular meristem before it can produce a flower is likely to be genetically determined, flowering may also be influenced epigenetically through the frond’s mother. A first daughter will already have its first daughter developing in the meristematic pouch, with at least one generation of internally developing fronds in *L. minor* and *S. polyrhiza* [64], and *W. microscopica* containing up to three unseen internal generations of daughter fronds nested within each other [10]. Evolutionarily conserved epigenomic stress responses such as DNA methylation and histone modification that influence flowering time in rice, sugar beets, *Arabidopsis*, and other plants [65,66], are likely to be active through these internal generations and perhaps future generations, and may help explain the long term effects of previous culture conditions on flowering rates. Comparing the results in early *Wolffia* screens (Appendix A) to experiments where inoculants came from fully acclimated cultures (Figure 5) underscores the importance of acclimating cultures for 4 weeks to fully eliminate variation before running experiments, as suggested in Duckweed Forum issue 8 [67].

### 3.1. Spirodela Polyrhiza

We focused on flowering of *S. polyrhiza* strains 7498 and 9509 because of the number of recent genomic and transcriptomic studies on them [11,29,30,34,35,37,38,68]. An abundance of molecular evidence illustrates the neotenous nature of *Spirodela* that must be overcome to induce flowering. Genome sequencing revealed a strong enrichment of genes for floral repressors, and a reduction in the number of members of the floral promoter gene families in both *Spirodela* strains compared to *Arabidopsis* [29,30]. About 9 members of the juvenile marker miR156 gene family were found, compared to 5 members of the adult marker miR172 gene family in the 7498 and 9509 genomes [29,30,68]. Sequencing of small RNAs in strain 7498 growing in 8 different stress conditions or hormonal applications, found that the ratio of miR156 to miR172 ranged from 71–408 [34]. These two highly conserved microRNAs regulate the abundance of the key floral promoters which integrate flowering responses to day length, stress, and accumulated sugar states [43,69,70,71].

Most strains of *S. polyrhiza* were classified as day neutral plants [1,72]. We found flowering in 12 h days, (Appendix A), although growth in CL produced the highest flowering rates (Figure 1). If inoculant was taken from 4-week-old cultures growing in CL in flasks containing 100 mL of Hg media, and used to inoculate Petri dishes containing 50 mL of Hg + 1.5 μM SA, we obtained optimal flowering of 11% for 7498 and 1% for 9509 (Figure 1). While both 7498 and 9509 produced pistils with stigmatic fluid and pollen capable of producing germination tubes (Figure 2 and Figure 4), gentle shaking, which was found to promote seed production in *L. gibba* G3 [16] failed to produce seed in *S. polyrhiza*. Our attempts to manually self-pollinate 7498 by touching dehiscent anthers to pistils, as done with *L. gibba* by Fu et al. [48] were also unsuccessful. 

Diverse types of stress can influence flowering in plants through SA signaling [73,74]. In *Spirodela,* a more common stress response than flowering and seed formation may be dormancy through the production of the starch-rich turions. In our optimal conditions, a 4 weeks old culture could have up to 39 mature and developing flowers, while at the same time having an average of 240 turions for strain 7498 and 134 turions for strain 9509. After two months, up to 500 turions could be found per flask of strain 7498, suggesting that genetic or environmental manipulation of turion production could dramatically influence the production of flowers, and vice versa. 

### 3.2. Wolffia Microscopica

*W. microscopica* flowers abundantly in the wild. These diminutive plants were used for many early studies on induction of flowering [75,76], and the morphology and anatomy of seed obtained from the wild have been studied [47]. In addition to being a promising crop plant which provides a complete protein [4], *Wolffia* is a prime candidate for molecular studies into how a tiny monocot manages to maintain both a vegetative and sexual reproductive cycle. The consistent laboratory protocol for maximum flowering induction we describe supports efforts to maximize research on, and use of, these plants. 

Our initial screen validated the reported abundant flowering of these plants, although there was a high degree of variability of replicates within a trial, and variability between replicated experiments (Appendix A). As with *Spirodela* and *Lemna*, variability in flowering was decreased when inoculants came from acclimated cultures (Figure 5). Culture inoculant size and age, EDDHA, and hormones like SA, are known to play roles in regulation of flowering of the Lemnaceae. We found that under our conditions, EDDHA had no effect on flowering rate (Appendix A), unlike what was reported earlier for a different strain of W. *microscopica* [77], or *S. polyrhiza* SP_20_ [78] and *L. gibba* G3 [79]. In addition to its chelating effects, EDDHA breaks down in a matter of weeks to form SA and SA-like molecules in sunlight [80], yet this transition from EDDHA to SA may not have occurred in our experiments where EDDHA was added to the medium after autoclaving. 

The differences we observed in flowering when plants were grown under the same conditions but in different culture vessels (Appendix A) suggests that a physical factor, such as rate of evaporation or accumulation of volatiles, surface to volume ratio, or even quorum sensing influences flowering in *Wolffia* and perhaps other duckweed genera. To test whether evaporation, or exchange of gases such as ethylene may be responsible, we tested flowering rate in Petri dishes and 6-well plates with and without parafilm. We found no significant difference due to parafilm, suggesting that surface to volume ratio or crowding may play a large role in flowering (Figure 5), although the mechanism for increased flower induction remains ambiguous, it may be SA independent.

### 3.3. Lemna

While gene editing and transformation present avenues for production of new duckweed varieties for basic and applied research, *L. gibba* G3 (strain DWC114) is currently the ideal strain for attempts to breed new varieties through sexual reproduction. This strain flowers readily, produces abundant dehiscent anthers, and self-pollinates to readily produce seed [16,48], (Figure 7 and Figure 10). Two different protocols to produce flowers and seed in *L. gibba* G3 have been described [16,48]. In the first, *L. gibba* G3 was grown on liquid E media in flasks in CL to produce flowers. Gentle shaking on a rotary shaker for 1 h twice a day was required to produce seed [16]. In the second protocol, the plants are grown in Petri dishes containing semi-solidified Modified Hoagland’s media supplemented with 20 μM SA in LD [48]. Flowers were either self or cross-pollinated manually under a stereoscope to produce seed [48]. It is possible that flowering under CL is influenced by a decrease in Circadian Clock Associated 1, which oscillates every 24 h under LD or SD, but is transcriptionally silenced when *L. gibba* G3 is grown under CL for 48 h or more [81]. Our best seed production protocol incorporated aspects of both previous protocols to both strongly induce flowering, and easily pollinate flowers with a gentle stir twice per week.

The need for supplementary SA to induce flowering if plants are grown in LD as found by Fu et al. [48] as compared to CL without SA [16] may also be explained if CL is acting as an abiotic stressor and thus increases endogenous levels of SA. Many plants respond to abiotic and biotic stress by the production of SA [73]. Intriguingly, *L. gibba* grown under CL exhibited increased auxin turnover [82]. Additionally, tight control of auxin levels is mediated through conjugation by GH3 acyl acid amido synthases, one of which, atGH3.5, has been shown to have a role in both auxin and SA homeostasis [83]. SA interacts with the flowering developmental pathway through Flowering Locus D. Flowering Locus D demethylates the histones associated with Flowering Locus C and thus inhibits transcription of floral promoters [84,85,86,87,88]. 

Fortunately, for a mechanistic understanding of floral regulation, we are no longer reliant on comparisons to other plant species, but can now analyze the transcriptome of flowering *L. gibba* G3 recently published by Fu et al., 2020 [89]. This study revealed differential expression of 1501 genes throughout the flowering process, and highlighted the role of photoperiod and SA. In LD *L. gibba* had very low levels of transcripts of two genes, *LgCO* and *LgGI*, which in *Arabidopsis* are responsible for integrating circadian rhythms. Fu et al. [89] found a lack of endogenous SA production, suggesting that flowering is being activated through exogenous SA from the medium. Surprisingly, this transcriptome suggested a flowering pathway distinctly different from other plant models, with a lack of expression of the floral regulators LgCO, LgGI, LgSOC1, and LgFD, and a co-expression of FT with LgTEM1 and LgSVP, which act as floral inhibitors in *Arabidopsis* [89]. Clearly there is a need for flowering transcriptome data from *Spirodela*, *Wolffia*, and the improperly developing *Lemna minor* strains, with which to build a larger picture of floral regulation strategies across this family.

Under optimal conditions for floral induction, *L gibba* DWC114 produced pistils with stigmatic fluid and both dehiscent and apparently non-dehiscent anthers, *L. minor* DWC130 produced pistils but no stigmatic fluid, *L. minor* DWC131 and *L. minor* DWC132 pistils produced stigmatic fluid but anthers remained non-dehiscent (Figure 8). Under our conditions, none of the *L. minor* strains produce seed, whereas *L. gibba* readily does. Over 70 attempts to manually fertilize *L. minor* DWC131 and DWC132 with *L. gibba* pollen were unsuccessful, indicating that the two species may be too diverged to form an interspecific hybrid (data not shown). However, the marked differences in floral development and fertility of these genetically related strains of *L. minor* and *L. gibba* provides an opportunity to identify critical genes in floral development. 

*L. gibba* seed can be left in a Petri dish to dry overnight, wrapped in parafilm, and stored dry at 4 °C for long-term storage. Seed can also be stored in diluted E medium, although light must be blocked to prevent germination. Of 234 seed stored in diluted E medium for 15 months, 36% germinated when placed in fresh E medium in the light. *L. gibba* seeds collected in the wild germinated at 70% efficiency after being stored in water for 2 years at room temperature, but germination dropped to 1% after 3 years of wet storage [90]. Only 5.6% of seed germinated after dry storage [90]. To eliminate possible fungal or bacterial contamination, seeds were surface sterilized with 5% bleach for 3 min or 10 min, then washed 3 times with sterile water, or more conveniently, rinsed with 70% ethanol and air dried. Ninety seven percent of one month old seed treated with any of the three methods germinated with no contamination. Single seed germinated within 4 days in as little as 5 mL of E medium in a 6-well plate, making it easy to rapidly obtain progeny from a cross.

Similar to many plant species that rely on protogyny (the sequential development of the pistil followed by the anther) to promote cross-fertilization while ensuring fruit production [91], like most family members the three genera of Lemnaceae we studied are protogynous [3,92,93,94]. Due to the clonal nature of Lemnaceae populations, protogyny would be substantially less effective at promoting cross-pollination and reducing deleterious in breeding than other self-incompatibility (SI) strategies [3]. Mechanisms involving the determinants at the S locus are diverse, and have evolved at least 35 times in the angiosperms, yet very little is known about them in monocots [95], and certain monocot orchids appear to have a novel SI mechanism [96]. Observations of flowering and seed production in the wild led to the suggestion that 22 species of duckweed are likely self-compatible [3]. Historical reports stated that *W. welwitschii* was able to self-pollinate [97], and that two *L. minor* populations were self-incompatible but capable of cross fertilization [98]. This work is summarized in Landolt and Kandeler (1987) [1]. Although Maheshwari (1956) [47] reported seed production by *W. microscopica* in the wild, it is not clear whether the wild populations were clonal or heterogeneous. We were unable to obtain seed from our *W. microscopica* and *S. polyrhiza* cultures even though pollen was capable of forming germination tubes, suggesting self-incompatibility in these species. It is therefore likely that SI is variable across the family, and co-cultivation of genotypes within a species may be required to develop breeding protocols. It will be interesting to investigate the relationship of SI mechanisms to genetic diversity in native habitats and the ability of a species to produce seed or develop turions. Both of these organs are capable of becoming dormant and surviving desiccation, and thus their formation might represent alternative stress-response strategies.

## 4. Summary

The Lemnaceae can be seen as new model plants ideally suited to probing important questions in plant biology, and for developing a new crop that can be a part of sustainably providing humanity with clean water, food, and fuel. To make future floral and breeding experiments easier we developed culturing protocols for obtaining flowering plants in the laboratory. 

In CL or LD, *L. gibba G3* flowers and sets seed in E medium supplemented with 20 μM SA. Occasional movement of the culture increases seed production.

*W*. *microscopica* flowers conveniently under LD in E medium in 6-well plates.

*S. polyrhiza* flowers in Petri dishes with 50 mL of Hg under CL when 1.5 μM SA is added.

Neither *Wolffia* or *Spirodela* produced seeds under these conditions, suggesting that self-incompatibility may be at play, and requiring further testing. With these culture conditions and modern molecular tools, it is now possible to determine the genetic mechanisms leading to flowering in *W. microscopica*, and determine if flowering in *S. polyrhiza* can reveal new insights in transposon mobility and DNA methylation. The Lemnaceae family is both exciting in its own right and an excellent basal monocot model with which to study the genetic mechanisms of floral regulation, or the mechanisms behind self-incompatibility in clonal populations. 

## 5. Materials and Methods

### 5.1. Plant Material and Culture Media

All strains of *S. polyrhiza*, *L. gibba*, *L. minor* and *W. microscopica* described are available from the RDSC (http://www.ruduckweed.org/ accessed on 26 January 2021). *W. microscopica* 2005 was kindly provided by Professor K.J. Appenroth, Friedrich Schiller University, Jena, Germany. DWC114, DWC130, DWC131, and DWC132 were barcoded at the chloroplast ATPase subunit I gene atpF-atpH intergenic spacer, according to the PCR protocol described in [28]. DWC132 was also barcoded using matK primers [28].

The media and culture vessels used for this study are described in Table 2 and Table 3. The chemical compositions of each of the media tested are provided in Appendix A, while the instructions for making stock solutions for, and preparing, E medium are in Appendix A. All media were supplemented with 29 mM sucrose unless specified otherwise with a minus (−) sign after the medium abbreviation. Additions of yeast extract and bactopeptone to E (Eye media) were made before autoclaving. SA was prepared as a 100 mM stock solution in ethanol. EDDHA was prepared as a 100 mM stock solution in water, and filter sterilized. SA and EDDHA were added to cooled media at the concentrations indicated. Flasks were fitted with cotton stoppers and loose aluminum foil covers. Petri dishes and 6-well plates were wrapped with strips of parafilm to maintain sterility and reduce evaporation.

All plants were grown at 24 °C, under General Electric Daylight 6500 K fluorescent bulbs or an LED panel. Plants in 16:8 h light:dark photoperiod (LD) were under a PPFD (Photosynthetic Photon Flux Density, μmol photons (400–700 nm)/m^2^/s) of 71–141, and a red (655–665 nm):far-red (725–735 nm) ratio of (2.54), while plants in continuous light (CL) or 12 h light:12 h dark days received a lower PPFD of 21–56, and a red:far-red ratio of (2.36). Supplemental far-red light experiments used custom red and blue LED panels providing PPFD of 362–516 and a red:far-red ratio of 1.69–1.84.

New cultures of *W. microscopica* were inoculated with a single 1.5 × 1.5 cm mesh loopful of floating fronds from a 3–5 week old mature culture growing in a 250 mL Erlenmeyer flask containing 100 mL E medium. *Spirodela* experiments were started with three 3–5 frond colonies from a 1, 2, or 4 week old culture grown in a flask of Hg medium. For *L. gibba* and *L. minor*, all experiments were started with 3 four-frond colonies from a 7 or 14 day old population growing in E medium in flasks.

### 5.2. Flowering, Seed Production and Seed Storage

Flowers were counted if they were developing or mature. An observable pistil, anthers, or a perfect flower (pistil with 2 anthers) were counted as one flower. Flowering rate is expressed as the percent of developing and mature flowers per 100 fronds. The number of fronds scored was either every frond in the culture vessel, or ≥100 fronds from any one culture. Flower or seed production data are presented as the average ± SEM (standard error of the mean, (*n* ≥ 3). Statistically significant difference between variables were calculated by Student’s *t*-test. Seed production is expressed as the total number per culture vessel. Seeds were stored dry, or in a flask of depleted media diluted 1:2 with sterile water at 4 °C. Unless otherwise specified, flowers were counted on days 7, 10, 14, 17, 21, 24, 28 after subculture.

### 5.3. Pollen Viability and Fertility

Pollen from all strains that produced anthers was tested for viability with a modified Alexander’s stain [52] and for the ability to form pollen tubes by gently spreading anthers on E solidified with agar. For staining, pollen from 5–10 anthers was stained, and at least 169 grains per anther were scored as viable or dead. Pollen tube formation was observed at 20× using a Leica DM550B microscope 1 h later. Fertility is described as the percent of pollen grains with germination tubes. At least 97 grains from 4–7 anthers were observed.

## Figures and Tables

**Figure 1 ijms-22-02733-f001:**
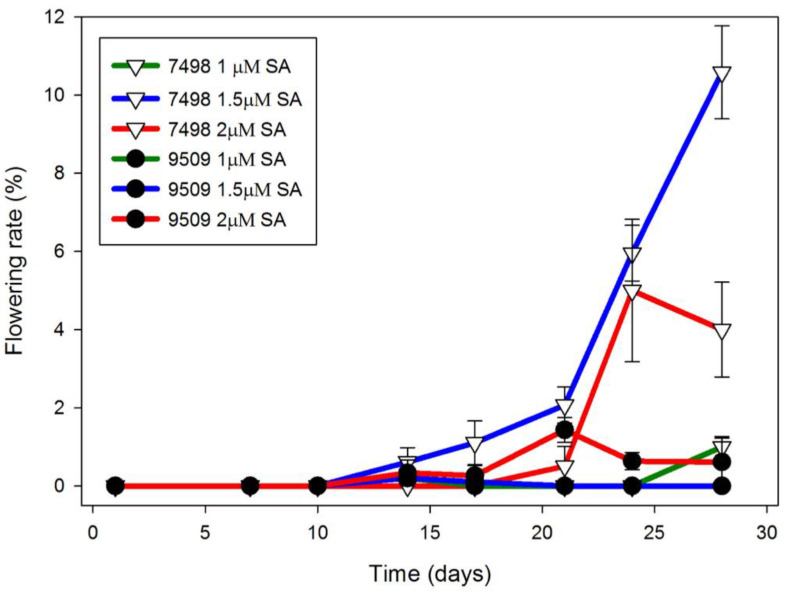
Flowering rate of *S. polyrhiza* strain 7498 cultured on Hg was increased to a much greater extent by SA than that of strain 9509. Plants were grown in Petri dishes inoculated with three 3-frond colonies from 4-week-old cultures growing on Hg without SA. Optimal flowering for strain 7498 was observed in the presence of 1.5 μM SA. Data are the average ± SEM (*n* = 3).

**Figure 2 ijms-22-02733-f002:**
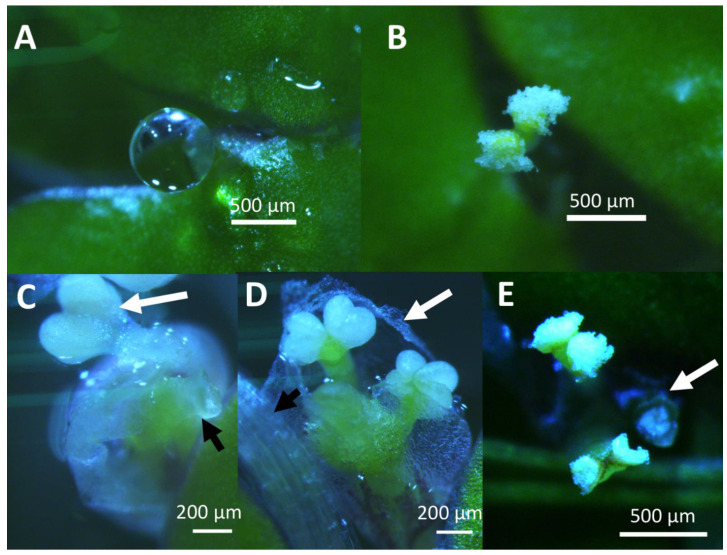
Flowers of *S. polyrhiza* strains 7498 and 9509. (**A**) Droplet of stigmatic fluid secreted by an *S. polyrhiza* 7498 pistil. (**B**) Dehiscing anthers of strain 7498. (**C**) Pistil without stigmatic fluid (black arrow) and non-dehiscent anther (white arrow) of *S. polyrhiza* strain 9509. (**D**) Two bi-lobed developing anthers and developing pistil of a strain 9509 flower. The transparent membrane known as a spathe (white arrow) has ruptured, but still surrounds the developing flower. The black arrow indicates a stipe, which connects the mother frond to a daughter. (**E**) Dehiscing anthers of strain 9509 and pistil without stigmatic fluid (white arrow).

**Figure 3 ijms-22-02733-f003:**
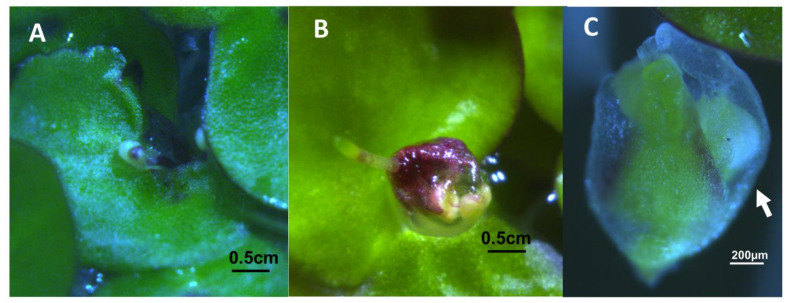
Anomalous flowers of *S, polyrhiza.* (**A**) An abnormal flower of strain 7498 that appears to have emerged from an abnormal, kidney shaped frond. (**B**) This inflorescence on strain 9509 appeared to be overgrown by the underside of the frond. The purple pigmentation is similar to that of the ventral surface of this strain. The vertical structure appears rhizoid-like. (**C**) An aborted inflorescence of strain 9509, consisting of the developing pistil and anthers enveloped in a spathe, found floating in the medium.

**Figure 4 ijms-22-02733-f004:**
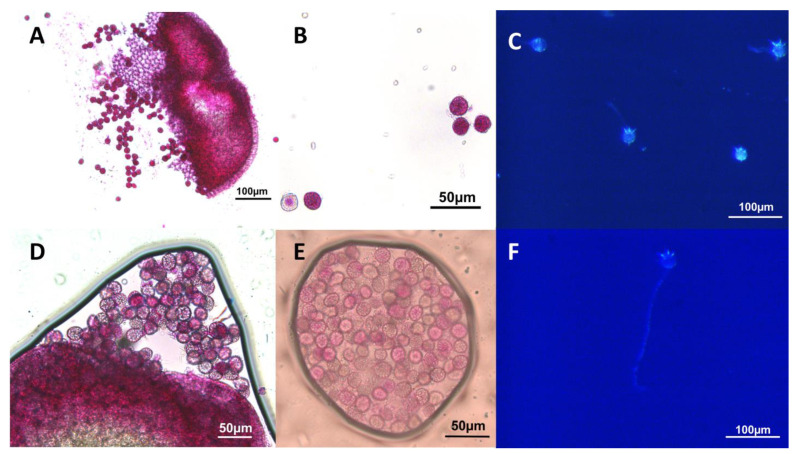
*S. polyrhiza* pollen analysis (**A**,**B**) Pollen from dehiscent anthers of strain 7498 stained with modified Alexander’s stain [52] indicating a high percentage of viability. Live pollen is pink or magenta, clear or blue stained pollen is aborted. (**C**) Pollen tube formation by pollen from dehisced anthers of strain 7498. (**D**,**E**) Alexander’s staining showed that anthers of strain 9509 produced viable pollen that was capable of forming pollen tubes (**F**).

**Figure 5 ijms-22-02733-f005:**
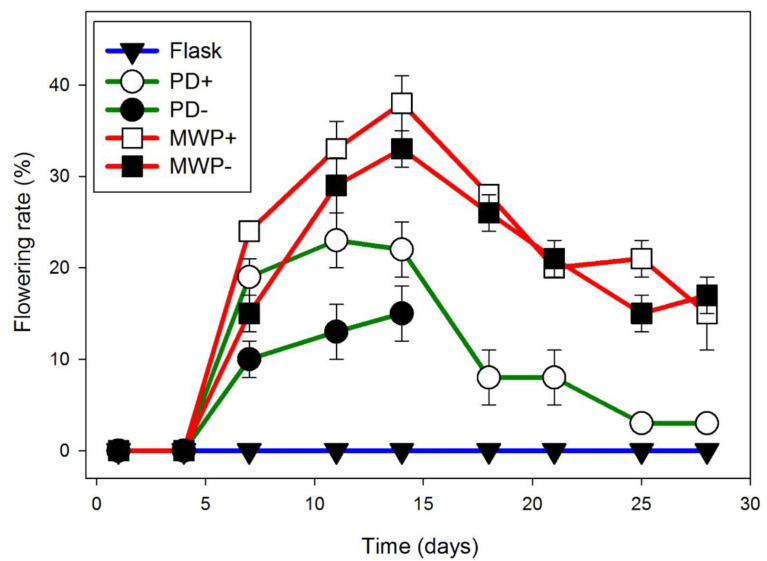
*W. microscopica* flowering rate is affected by culture vessel. Flowering rates of *W. microscopica* growing on E media in cotton stoppered flasks (blue), Petri dishes (PD) (green), or multi-well plates (MWP) (red). Petri dishes or 6-well plates were sealed with parafilm (open symbols) or left unsealed (closed symbols). Maximal flowering was observed at day 15 after inoculation in sealed 6-well plates. Data are the average ± SEM (*n* = 3).

**Figure 6 ijms-22-02733-f006:**
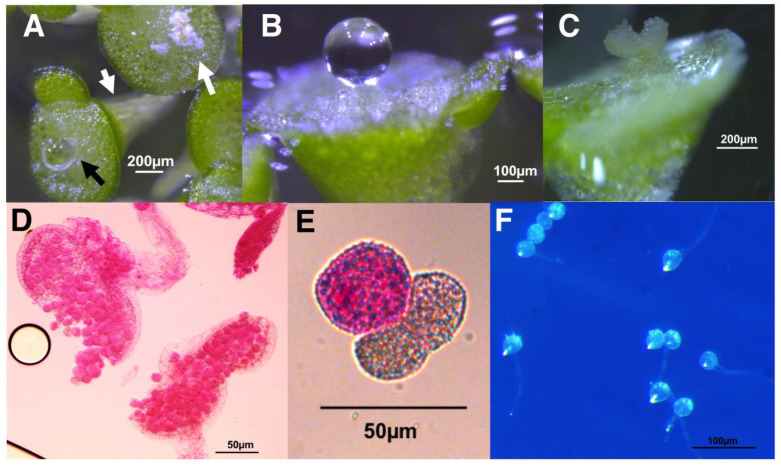
*W. microscopica* 2005 flowering and pollen analysis (**A**) Aerial view of mature *W. microscopica* 2005 fronds showing the pseudoroot (white arrowhead) extending from the ventral surface of a frond that is asexually reproducing a daughter frond at the same time it has produced a pistil. The pistil emerged from a furrow on the dorsal surface and has secreted stigmatic fluid (black arrow). A dehiscing anther has emerged from the furrow of a different frond (white arrow). (**B**) Side view of mature pistil with secreted fluid. (**C**) Side view of a dehiscing bilobed anther. Morphology is as described in Sree et al. [10]. Ruptured anthers releasing pollen (**D**) and pollen grains (**E**) stained with modified Alexander’s stain indicating that *W. microscopica* 2005 is producing viable pollen in culture. (**F**) 28% of *W. microscopica* 2005 pollen produced pollen tubes.

**Figure 7 ijms-22-02733-f007:**
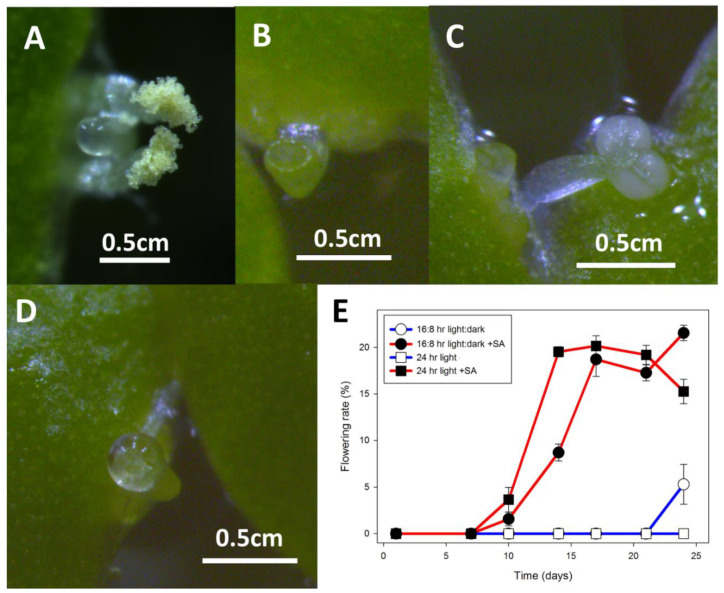
Flowering of *L, gibba* G3 and *L. minor* strains growing on E medium supplemented with 20 μM SA. (**A**) *L. gibba* DWC114 flower with the pistil having secreted a drop of clear stigmatic fluid and pollen on both dehiscent anthers. (**B**) *L. minor* DWC130 pistil with no stigmatic fluid (**C**) *L. minor* DWC131 pistil and mature non-dehiscent anther. (**D**) *L. minor* DWC132 pistil with a drop of stigmatic fluid. (**E**) Flowering of *L. gibba* G3 DWC114 on E supplemented with 20 μM SA is independent of LD or CL lighting conditions. Data are the average ± SEM (*n* = 3).

**Figure 8 ijms-22-02733-f008:**
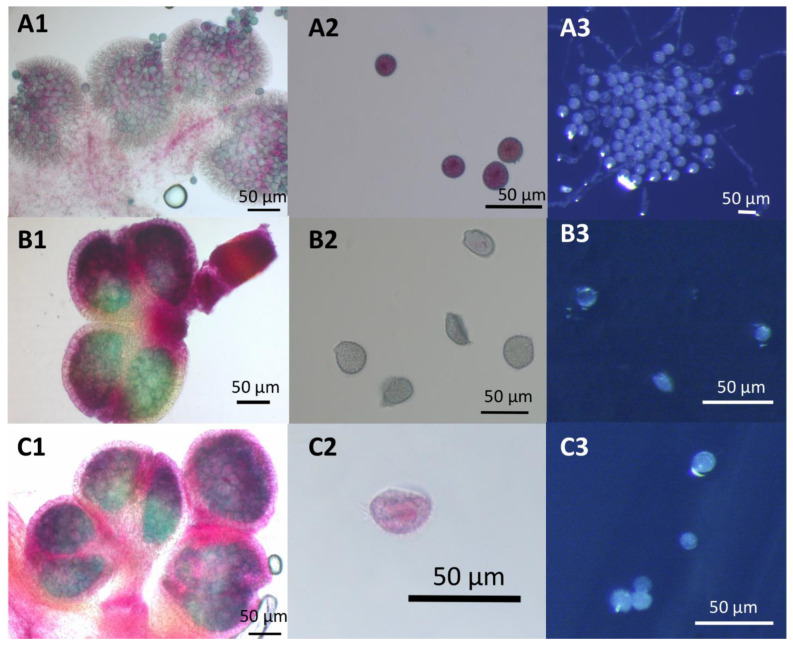
Pollen viability and germination assays for *Lemna* DWC114, DWC131, and DWC132 (**A**) *L. gibba* G3 DWC114 anthers and pollen (**B**) *L. minor* DWC131 anthers and pollen (**C**) *L. minor* DWC132 anthers and pollen (1 and 2) Anthers and pollen were stained with modified Alexander’s stain. (3) Pollen tube formation after 1 h of germination on E media.

**Figure 9 ijms-22-02733-f009:**
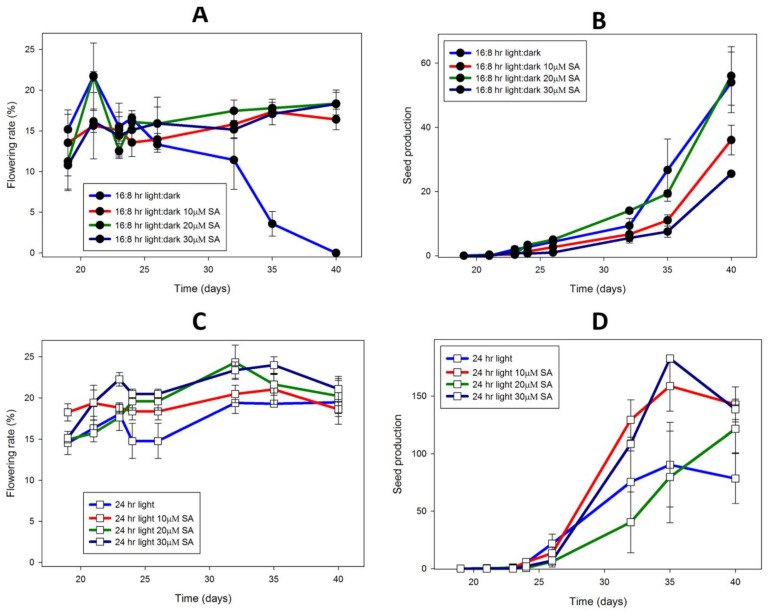
*L. gibba* G3 DWC114 flowering is not affected by daylength but requires SA, whereas seed production is greatly increased under constant light, and is dependent on SA concentration. Flowering rate is described as the percent of flowering fronds per 100 fronds. Seed production was measured as the number of seed per flask. Flowering (**A**) and seed production (**B**) In LD with various concentrations of SA. Flowering (**C**) and seed production (**D**) under CL with various concentrations of SA. Data are the average ± SEM (*n* = 3).

**Figure 10 ijms-22-02733-f010:**
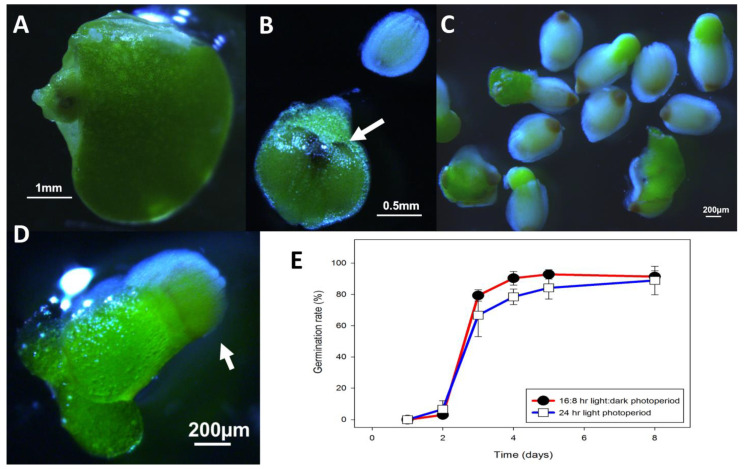
*L. gibba* G3 DWC114 seed development and germination (**A**) Developing seed growing in the meristematic pouch, (**B**) Detached fruit with 2 developing seed. Note the two hyperpigmented spots, which are the opercula (white arrow). The operculum is visible on the free seed, which shows the typical ribbing on the seed coat. (**C**) Germinating seed showing emergence of the new green frond together with ungerminated seed at the bottom of the culture vessel. (**D**) A seedling floating on the surface of the medium still attached to the seed coat (white arrow), and already undergoing asexual budding. (**E**) Germination percent of seed over eight days in E in multi-well plates in LD or CL lighting conditions. Data are the average ± SEM (*n* = 3).

**Table 1 ijms-22-02733-t001:** DNA Barcoding results showed that only DWC114 is *L. gibba* and that DWC130, DWC131, and DWC132 are *L. minor*. DNA barcoding results are given as the strain identification (genus and species, with ID in parentheses) and accession numbers for the first and second matches from BLAST [51] searches with the chloroplast atpH-atpF intergenic region from each strain.

Strain	1st Match	Accession	2nd Match	Accession
DWC114	*Lemna gibba* (5504)	KX212889.1	*Lemna gibba* (7741)	KX212887.1
DWC130	*Lemna minor* (7210)	KX212888.1	*Lemna minor*	DQ400350.1
DWC131	*Lemna minor* (7210)	KX212888.1	*Lemna minor*	DQ400350.1
DWC132	*Lemna japonica* (0216)	KJ921747.1	*Lemna minor* (7210)	KX212888.1

**Table 2 ijms-22-02733-t002:** Media.

Media	Abbreviation	Reference	pH	Supplier
E ^a^	E	[60,99]	4.6	Chemicals from Sigma-Aldrich, St. Louis, MO, USA
Hoagland’s ^a^	Hg	[100]	5.8	Cassion Labs, Smithfield, UT, USA
Shenk Hildebrandt ^a^	SH	[101]	5.8	Sigma-Aldrich

^a^ The chemical composition of each of these media along with the other common medium, Steinberg [102], can be found in Appendix A. All media were adjusted for pH with KOH, supplemented with 29 mM sucrose, and autoclaved. All flasks had cotton stoppers with loose aluminum foil covers unless stated otherwise.

**Table 3 ijms-22-02733-t003:** Containers.

Container	Container Size	Volume of Media	Supplier
Flask	250 mL	100 mL	VWR, Bridgeport, PA, USA
Small flask	125 mL	50 mL	VWR
Petri dish	100 × 15 mm	50 mL	Kord-Valmark, Bridgeport PA, USA
Multi-well Plate	6 well	10 mL	Corning, NY, USA

## Data Availability

All data presented in article and Appendix A.

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
