# Peer review of "Flowering and Seed Production across the Lemnaceae"

_ijms, 2021, doi:10.3390/ijms22052733_

Round 1
Reviewer 1 Report
The manuscript describes a protocol for inducing flowering in three duckweed genera. The introduction does a good job in describing the importance and timeliness of this kind of research. The methods are clearly presented, except for statistical analysis; the data presented in the main text have not been subjected to statistical scrutiny. The legend of Figure S3mentions (line #849) that the data has been statistically analysed, although the figure itself does not include any notions on statistical analysis. The methods possibly used for statistical analysis are not described. Because the authors state that the variance between replicates was high, showing results from statistical analyses would be important for interpretation of the data.
The results chapter is clear and nicely illustrated with photographs, and the discussion deals with the results at an adequate depth.
Minor notes:
Lines #446-449: are there some words missing?
Lines #501-503: an unclear sentence. Please clarify.
All in all, I see that this manuscript provides an important contribution to developing the duckweed genera as model species.
Author Response
- We included our statistical analysis in the methods section at line 673-675
“Flower or seed production data are presented as the average ± SEM (standard error of the mean, (n³3). Statistically significant difference between variables were calculated by Student’s t-test.”
- For Figure S3, we clarified the figure legend lines 869-872 below, and revised the figure with the *s.
Figure S3. Flowering rate and seed production in strain DWC114
(A) Flowering rate in Eye media. Data are the average ± SEM (n=3). * indicates a significant difference (p<0.05), ** indicates p<0.01, between the two conditions (n=3) according to Student’s t-test.
(B) Total number of seed produced, with the same statistical analysis.
- We formally analyzed the differences of petri dishes and 6 well plates, as well as parafilm to better understand these variables on Wolffia Lines 252-255
“After day 7, flowering was significantly greater in parafilm sealed 6-well plates (red) than in parafilm sealed petri dishes (green) (p<0.05) and no significant differences between 6-well plates sealed or not sealed with parafilm (Figure 5).”
- Looking at Figure 1, and the underlying data, we included the following comments on statistically significant results. Lines 173-176
“By day 24, strain 7498 had a significantly higher flowering rate than strain 9509 the 1.5 mM SA SA condition (p<0.01). By day 28, when the culture filled the dish, 11 ±1% of the 7498 fronds growing on media containing 1.5 mM SA were flowering, significantly different from 2.0 mM SA (p<0.01).”
- We statistically quantified one of the differences in Figure 7e. Lines 323-327
“The results show that the presence of SA was required to produce flowering fronds until very late in the culture cycle. In the presence of SA, flowering under CL was significantly greater than in LD at day 14 (p<0.01) but the appearance of flowers was essentially equal thereafter under either light condition until the end of the culture cycle.”
Reviewer 2 Report
Very interesting work. I have no comments!
Author Response
Dear Reviewer #2,
Thank you for taking the time to review and evaluate the scientific quality of our manuscript. Thanks for your approval and interest in the study, and we sincerely hope it pushes this field forward.
We thank you for your prompt scientific evaluation of this manuscript and helpful guidance. According to the suggestions of yourself, and Reviewer #2 we’ve performed several minor edits, highlighted in the revised manuscript, and appreciate your suggestions in improving the quality of the paper.
At lines 467-469 we made the following changes to make the sentence easier to understand.
“If inoculant was taken from 4 week old cultures growing in CL, in flasks containing 100 ml of Hg media and used to inoculate petri dishes containing 50 ml of Hg ± 1.5 mM SA, we obtained optimal flowering of 11% for 7498 and 1% for 9509 (Fig. 1).”
At lines 523-534 we clarified the section as follows.
“Two different protocols to produce flowers and seed in L. gibba G3 have been described [16, 48]. In the first, L. gibba G3 was grown on liquid E media in flasks in CL to produce flowers. Gentle shaking on a rotary shaker for 1 hour twice a day was required to produce seed [16]. In the second protocol, the plants are grown in petri dishes containing semi-solidified Modified Hoagland’s media supplemented with 20 mM SA in LD [48]. Flowers were either self- or cross-pollinated manually under a stereoscope to produce seed [48]. It is possible that flowering under CL is influenced by a decrease in Circadian Clock Associated 1, which oscillates every 24 hours under LD or SD, but is transcriptionally silenced when L. gibba G3 is grown under CL for 48 hours or more [81]. Our best seed production protocol incorporated aspects of both previous protocols to both strongly induce flowering, and easily pollinate flowers with a gentle stir twice per week.”
Recently, we’ve worked with our colleagues at the RDSC, to help them in updating the stock collection to reflect our findings.
We therefore changed lines 296-298 from
“The barcoding results were reported to the RDSC to ensure the accuracy of the global collection”
To
“Based on the results our colleagues at the RDSC, updated the database as part of their curation of the global collection”